# Perception of healthcare administrators on the impediments of optimizing adverse events following immunization e-Reporting in Nigeria

Grace Fubara Erekosima[1], Saheed Dipo Isiaka [2]*, Folake Oni[3], Ahmed Rufai Garba[4], Oluchi Bassey[1], Stephen Olabode Asaolu[2], Olugbemisola Wuraola Samuel[2], Genevieve Ozioko[1], Oluwafisayo Azeez Ayodeji[1], Euphemia Chigekwu Agomuo[1], Irene Odira Okoye[1], Sidney Sampson[1,2], Precious Iyayi[1], Victor Daniel[5]

**1** Sydani Initiative for International Development, FCT, Abuja, Nigeria, **2** Sydani Institute for Research and Innovation, FCT, Abuja, Nigeria, **3** Renewed Minds, FCT, Abuja, Nigeria, **4** National Primary Healthcare Development Agency, FCT, Abuja, Nigeria, **5** Acasus Nigeria, FCT, Abuja, Nigeria

* saheed.isiaka@sydani.org

**Data availability statement:** This manuscript is a sub-set of a landscape assessment conducted to optimize the AEFI surveillance and reporting system in Nigeria through a collaborative partnership among the Federal Ministry of Health, the National Primary Healthcare

## Abstract

### Background

Adverse events following immunization (AEFI) are any negative medical event that occurs after vaccination but may or may not be causally related to the vaccine. AEFI reporting is the gateway to AEFI surveillance systems at primary healthcare facilities where immunization services are provided. Several studies have highlighted low reporting of AEFI cases, particularly in low-resource settings, yet nothing is known about stakeholders' perspectives on the factors associated with the AEFI reporting rate in Nigeria.

### Objective

This study explored the stakeholders' perspectives (from a baseline assessment) on the barriers to adequately reporting (electronically) AEFI in Nigeria.

### Methods

The study was conducted using a qualitative approach. Key informant interviews were conducted at the national and sub-national (state) levels, who were purposively selected to acquire information from stakeholders on the challenges facing the e-reporting of AEFI at the national and state levels. All the audio interview files were transcribed into English Language, coded, and presented using a thematic approach.

### Results

A total of 32 healthcare workers at the national and sub-national levels participated in the study. The study adapted an extant pharmacovigilance thematized

Development Agency, and Sydani Group, and funded by Bill and Melinda Gates Foundation (BMGF). Before the landscape assessment, an agreement was made with the Federal Ministry of Health, which is also the Institutional Review Board (IRB) custodian for all health-related research activities in Nigeria. The agreement includes a non-disclosure of facility-related information, which are heavily contained in the transcripts from the assessment. This is also why we could only write from a section of the assessment (because we believe it will be relevant to other developing countries). Providing the data as an open access will be considered a threat to national security by the two government health authorities in Nigeria. Hence, the transcripts will be provided on a case-by-case request basis. Additionally, the funding body also highlighted in the project contract that all data obtained from the project are proprietary of the national health coordinating body and should only be utilized by our organization or any other third party upon agreement with the coordinating body. The contact addresses that can be reached should the data be required is as follows: Organization name: Sydani Institute for Research and Innovation Contact Person: Saheed Dipo Isiaka (email: saheed.isiaka@sydani.org) Contact Person 2: Grace Erokosima (email: grace.erokosima@sydani.org) Contact person 3: Genevieve Ozioko (email: Genevieve.ozioko@sydani.org) Contact person 4: Queeneth Chigozie (email: queenethguzie@gmail.com) - representative of the National Primary Health Care Development Agency (NPHCDA).

**Funding:** The author(s) received no specific funding for this work.

**Competing interests:** The authors have declared that no competing interests exist.

framework of reporting the barriers to electronic reporting of the pharmacovigilance system. Therefore, this study categorized the participants' responses into four main themes, including healthcare workers' knowledge deficiency and fear, technical infrastructural challenges, poor reporting systems, and inconsistency in government commitment.

## Conclusion

This study concludes that the AEFI surveillance system in Nigeria requires immediate and thorough attention. This stems from evidence gathered from the study participants, revealing the various challenges that are extant at the national and subnational levels. While these challenges – healthcare workers' knowledge deficiency and fear, poor technical infrastructure, poor reporting system, and inconsistency in government commitment – may appear mundane, they are critical to optimizing the AEFI surveillance system and maintaining the drive for an improved disease management system.

## Recommendation

This study recommends that stakeholders at all levels should take up improved ownership of AEFI reporting (especially electronic) systems in the country.

## Background

Immunization is a highly cost-effective intervention that protects against preventable diseases and reduces morbidity and mortality [1–4]. Globally, immunization prevents an estimated 4–5 million fatalities across all age categories, from diphtheria, pertussis, tetanus, and measles each year [5,6]. Despite significant progress in vaccinations and prevention against vaccine-preventable diseases, adverse events following immunization (AEFI) occasionally occur, which can undermine public confidence in immunization programs.

Adverse events following immunization refer to any negative medical event that occurs after vaccination but may or may not be causally related to the vaccine [7]. It may be non-serious, resolving without long-term effects [8,9], or serious, leading to hospitalization or death [10]. AEFI has been highlighted as one factor contributing to Nigeria's low vaccination rates [11]. Due to the potential of AEFIs to stem or reverse the gains in vaccination programs, AEFI surveillance systems have therefore become an integral part of national immunization systems [12].

AEFI reporting is the gateway to AEFI surveillance systems and usually takes place at primary healthcare facilities where immunization services are provided, making healthcare workers the primary point of contact for clients (and in some cases the caregivers) reporting AEFI [13]. AEFI reporting rates are usually low in developing countries despite the important role of AEFI reporting, and this falls below the WHO reporting standard of 10 AEFIs per 100,000 surviving infants [14]. In Nigeria, limited

data exist on AEFI reporting rates for childhood immunization in selected states, and no national aggregate data exist from the literature. According to Ogundele and Colleagues [15], the current reporting rate of AEFI varies across different states in Nigeria, and ranges from 19.3% to 57%.

Some of the factors that have been reported outside Nigeria to be responsible for low reporting of AEFI include health-care workers' turnover, lack of knowledge of the reporting options, lack of information on reportable AEFIs and reporting structures, misunderstanding of reportable AEFIs, complicated form of reporting, time constraints, and fear of raising false alarms about a vaccine [16–19]. Omoleke et al [20] identified a shortage of human resources, knowledge gaps, limited training, excessive workload, caregiver's factor, governance, and leadership – moribund AEFI committee, and oversight and weak implementation of AEFI policy guidance, as major barriers affecting the functionality of the AEFI surveillance system in Nigeria, including reporting.

There are two main channels for AEFI reporting in Nigeria: health facility reporting using both paper and electronic reporting tools (District Health Information System 2 - DHIS2 app) and direct client reporting using the Med Safety app. The Med Safety App was introduced by the National Agency for Food and Drug Administration and Control (NAFDAC) to report cases of adverse drug reactions. At the facility level, AEFIs are documented using paper-based or e-reporting tools. The process of paper-based AEFI reporting entails clients reporting their experiences to the health facilities, where they are documented by the Routine Immunization Officer (RIO) or the Officer-in-Charge (OIC). All the AEFIs in the health facility are then collated and submitted to the Disease Surveillance and Notification Officer (DSNO) at the local government level. Subsequently, the DSNO at the state level receives all collated AEFIs from local government area DSNOs and submits the compiled data to the National Primary Health Care Development Agency, through the state DSNO and/or Monitoring and Evaluation (M&E) Officer.

Prior to the advent of the COVID-19 pandemic, and the subsequent mass vaccination exercises that ensued, the system of reporting AEFI at the health facility level was paper-based, which was synonymously fraught with challenges of data quality in terms of reliability, timeliness, availability, and completeness of reporting. Owing to this, the introduction of the COVID-19 vaccines prompted the development of an AEFI module on the national health information management system's District Health Information Software 2 (DHIS-2), developed and introduced by the Health Information System Program (HISP) center, University of Oslo, Norway [21]. This resulted in the emergence of reporting AEFIs in a dual mode, using both the extant paper-based approach and the newly introduced electronic reporting at various health facilities across the country.

The primary goal of integrating the AEFI module into DHIS-2 is to ensure real-time visualization of reported AEFI events by all stakeholders across various levels for decision-making. Routine spot-checks revealed gaps in the capacity of healthcare workers to optimize the utilization of the e-reporting channels. To address this gap and improve the AEFI reporting system in Nigeria, a baseline assessment was conducted to explore stakeholders' perspectives and document other confounding challenges that contribute to the barriers of reporting (with emphasis on electronic reporting) AEFI events in Nigeria. This article aims to report our findings.

## Methodology

### Study design

This study was anchored on an exploratory research design to acquire qualitative data from purposively selected participants in Nigeria. Key informant interviews were conducted at the national and sub-national (state) levels to understand the perspectives of AEFI surveillance stakeholders on the challenges of e-reporting at the national and state levels and the ways by which e-reporting can be optimized. Participants were recruited and interviewed between September 12 and October 27, 2022. The recruitment period took about seven weeks due to competing priorities on the part of the health administrators, particularly at the national level.

## Study settings

The study was conducted in twelve (12) states across the six geopolitical zones in Nigeria at the sub-national level, and the federal capital territory at the national level. Nigeria is a multinational country with almost 230 million people, spread across 250 ethnic groups, speaking 500 different languages and identifying with a wide range of cultural lives. Nigeria is the sixth most populous country in the world and the most populated country in Africa. Nigeria is divided into six geopolitical zones, including Northwest, north-central, northeast, south-south, southwest, and southeast.

One of Nigeria's six geopolitical zones, which collectively make up the Middle Belt, is the North Central one. It consists of Nigeria's Federal Capital Territory as well as the six states of Benue, Kogi, Kwara, Nasarawa, Niger, and Plateau. From the border with Cameroon to the border with Benin, the North Central region spans the whole breadth of the nation. Approximately 20 million people live in the region, making up 11% of the nation's overall population.

Geographically, the Northeast represents the largest geopolitical zone in the country, making up almost one-third of all of Nigeria. It is made up of the states of Taraba, Yobe, Borno, Bauchi, Adamawa, and Gombe. With a population of over 26 million, the region makes up 12% of the nation's total population. It is well-known for its cattle and its abundant crop development, both of which have a significant impact on the national economy. When compared to the country's southern portion, the area is not as heavily populated.

Nigeria's northwest is referred to as the Northwest both politically and geographically. It is made up of the following seven states: Zamfara, Jigawa, Kaduna, Kano, Katsina, Kebbi, and Sokoto. Culturally speaking, the majority of the zone is contained inside Hausaland, the ancestral territory of the Hausa people, who comprise the majority of the population in the northwest. Significant minorities of Fulani people and other ethnic groups do exist, mostly on the zone's edges.

The Southeast, which includes the five states of Abia, Anambra, Ebonyi, Enugu, and Imo, also denotes the political and geographic region of inland southeast Nigeria. It is the smallest geopolitical zone, but it makes a significant economic contribution to the country because the majority of its territory is Igboland, the indigenous cultural homeland of the Igbo people, who make up the majority of the ethnic population in the southeast.

The six states that make up the South-South are Rivers, Edo, Delta, Cross River, Akwa Ibom, and Bayelsa. A large portion of the Niger Delta, which is essential to the region's ecology and economic growth, is enclosed by the zone. Despite making up just around 5% of Nigeria's total area, the South-South's substantial oil and natural gas deposits enable it to make a significant economic contribution to the country. With over 26 million residents, the zone makes up 12% of the nation's total population.

The states of Ekiti, Lagos, Ogun, Ondo, Osun, and Oyo make up the Southwest. The majority of the zone is culturally part of Yorubaland, the native country of the Yoruba people, who comprise the largest ethnic group in the southwest. The Yoruba kingdom also extends in part to some areas in Kwara and Kogi states, and together they make up the Nigerian Yorubaland. Nigeria's economy benefits considerably from the economic contributions of the Southwest's urban centers, primarily Lagos and Ibadan. Over 50 million people live in the region, slightly over one-fifth of the nation's population. Lagos is the most populated city in Africa, the Southwest, and Nigeria (Please see Table 1).

The study participants are health administrators within the primary healthcare system and include those at the national and sub-national levels. The health administrators at the state level are particularly incorporated into the Nigerian Centre for Disease Control (NCDC) and are often members of the disease outbreak response team at the sub-national or state level. The health administrators at the national level include the director and deputy director of disease control and immunization. The healthcare workers at the state level, participants include the State epidemiologist, the state disease surveillance and notification officer (DSNO), the state Immunization Officer (SIO), the State monitoring and evaluation (M&E) officer, and the State Health Education Officer (SHEO)

However, this study does not include the category of healthcare workers who directly report AEFI at the facility level. The categories of healthcare workers who are often faced with this responsibility are the routine immunization officers

**Table 1. State selection across the geo-political zones of Nigeria.**

| Level | Geo-Political Zones | States |
|---|---|---|
| National | | Federal Capital Territory |
| Sub-National | North-central | Benue and Nassarawa |
| | Northeast | Bauchi and Taraba |
| | Northwest | Jigawa and Kano |
| | Southeast | Anambra and Enugu |
| | South-south | Bayelsa and Edo |
| | Southwest | Ogun and Osun |

(RIOs), recorders, and, in a few cases, nurses in the facilities (as a backstop for RIOs). If recorded manually, data is often submitted to the local government area Monitoring and Evaluation Officer (LGA-M&E), who subsequently reports to the state level, and that is eventually submitted to DHIS2 by the state M&E officer. Electronic reporting often involves direct data uploading into DHIS2 and is verified by the state M&E officer

## Sampling

For this study, state-level participants were purposively selected from 12 states, plus the purposive selection of the Federal Capital Territory, covering the 6 geopolitical zones. Two states were chosen from each geopolitical zone based on

(i)   High/low reporting of COVID-19 AEFIs

(ii)  Availability of uninterrupted network services

## Inclusion and exclusion criteria for study participants

The participants at the sub-national level were purposively identified and selected based on their job positions and roles in the AEFI surveillance system in their respective states. At the national level, members of the AEFI technical working group (TWG) who were also involved in coordinating disease surveillance at the state levels were selected for the interview (based on availability and consent). On the other hand, participants were excluded at the sub-national level if their job titles did not fit into those responsible for disease surveillance. Also, participants who were not involved in the technical working group at the national level were excluded.

   The DSNO, State Epidemiologist, state M&E officer, and SIO made up the subnational stakeholders. There were two national stakeholders interviewed. Six (6) of the intended thirty-six (36) participants at the sub-national levels were absent at the time of the study due to certain constraints (Please see Table 2).

## Ethical approval and consent to participate

The study adhered to the ethical guidelines outlined in the Helsinki Declaration for research involving human subjects. Prior to the key informant interviews (KIIs), participants provided both written and verbal consent. Strict measures were implemented to ensure confidentiality of information, and the discussions were conducted anonymously. Furthermore, the study protocol received approval from the National Health Research Ethics Committee (NHREC) of the Federal Ministry of Health (FMoH) in Abuja, with approval number NHREC/01/01/2007–19/08/2022.

## Data collection

A semi-structured interview guide, based on the study objective, was developed for the study respondents to generate the data needed for this study. Participants were invited for interview sessions via electronic mail and a letter of introduction

**Table 2. Participants Selection across the study locations.**

| S/N | Participants | IDI | Total |
|---|---|---|---|
| **National Level Administrators** | | | |
| 1 | Director for Disease Control and Immunization | 1 | 1 |
| 2 | Deputy Director for Disease Control and Immunization | 1 | 1 |
| **State Level Actors** | | | |
| 3 | State Immunization Officer | 1*8 | 8 |
| 4 | State Disease Surveillance and Notification Officer | 1*12 | 12 |
| 5 | State epidemiologist | 1*10 | 10 |
| | Total number of participants across all levels | | 32 |

from the National AEFI TWG of the National Primary Healthcare Development Agency (NPHCDA). Physical interview sessions were held with respondents at the national level, while respondents across the selected states were interviewed virtually through the Zoom platform. Specifically, the study assessed participants' perceptions of the challenges facing healthcare workers in the process of reporting AEFIs. This was focused on the need to understand and gather insights into the reality of the AEFI reporting system in Nigeria. The findings from the acquired data were used to design tailored interventions to strengthen the AEFI surveillance system thereafter. Participants were asked questions such as "Tell me briefly about yourself, your work, and your role in the AEFI Surveillance structure. In your opinion, do you think it is possible to integrate AEFI surveillance with other existing state health programs? What are your thoughts on the current AEFI data electronic reporting methodology? And What are your thoughts on the utility of the AEFI electronic data management system?".

## Data collection procedure

A total of thirty (30) interviews were conducted at the sub-national level, while two (2) interviews were conducted at the national level. Written Consent (for national level participants), and Verbal consent (for sub-national level participants) were obtained from the respondents before all the interviews were conducted. All interviews were conducted in English Language to facilitate easy communication. The interviews, which lasted between 18 and 23 minutes, were recorded and stored on a local audio device.

## Data analysis

All recorded interviews were conducted in English, and the recorded files were transcribed verbatim in English language by professional transcribers. Transcripts from interviews were triangulated with notes taken by data assistants during data collection to ensure that the transcripts accurately captured all points.

A hybrid approach (inductive and deductive) to thematic analysis, described by Swain [22] was used to analyze the transcripts. Firstly, a codebook was developed using a framework of barriers to electronic pharmacovigilance systems defined by Agoro *et al* [23]. The Pharmacovigilance system is the scientific interpretation of detecting, assessing, understanding, and preventing adverse drug or vaccine events (post-usage). Following transcription, two members of the research team read through the transcripts to identify emerging themes, and the codebook was then refined based on emerging themes from the transcripts. Using the revised codebook, all transcripts were coded deductively using the Dedoose software (Version 9) by a team of 4 professional coders. Data synthesis was conducted thematically based on the pattern identified in the participants' responses. Four thematic and sub-thematic areas were generated from the observed pattern of responses.

## Results

### Socio-demographics of the study participants

A total of 32 respondents were interviewed. Two respondents were officials from the NPHCDA, and 30 were members of the state AEFI committee. No respondent withdrew from the study.

Following the interview sessions held with the study participants, the study adapted Agoro *et al* [23] thematized framework of reporting the barriers to electronic reporting of the pharmacovigilance system. Therefore, this study developed and categorized the participants' responses into **four main themes** (please see Fig 1). The thematic areas are (i) healthcare workers' knowledge deficiency and fear, (ii) technical infrastructural challenges, (iii) poor reporting systems, and (iv) inconsistency in government commitment. Responses were discussed across various perspectives using the participants' expressions on various issues that contributed to the challenges that apply to reporting AEFI on COVID-19 vaccination and routine immunization in Nigeria.

### Healthcare workers' knowledge deficiency and fear

**Knowledge deficiency among healthcare workers.** Findings from the study revealed the existence of a knowledge gap among healthcare workers. Essentially, this knowledge gap indicates a "*clog in the wheel*" for the operational capacity of healthcare workers in various primary healthcare facilities regarding AEFI in the immunization field, and as such, would require knowledge acquisition through training workshops. This is evident in the quote below, demonstrated by one of the study's participants.

*The healthcare workers do not really understand the importance of reporting, and that is why they do not know about it, they should learn the importance of this, and they should maybe once in a while be engaged in training so that they will know what is needed to be done on the field then there should be supervision once in a while, to see what they should report, why they should report, whom to report to or if they are not reporting, why are they not reporting and this will also allow us to know if they have the tools that they are using for reporting (KII/ State Level/2022)*

**Drivers of knowledge deficiency among healthcare workers.** Findings from our study have revealed that some factors have been reported to contribute to the lack of knowledge on AEFI reporting within health facilities, and these play major roles in the misunderstanding of identifying and reporting AEFI cases.

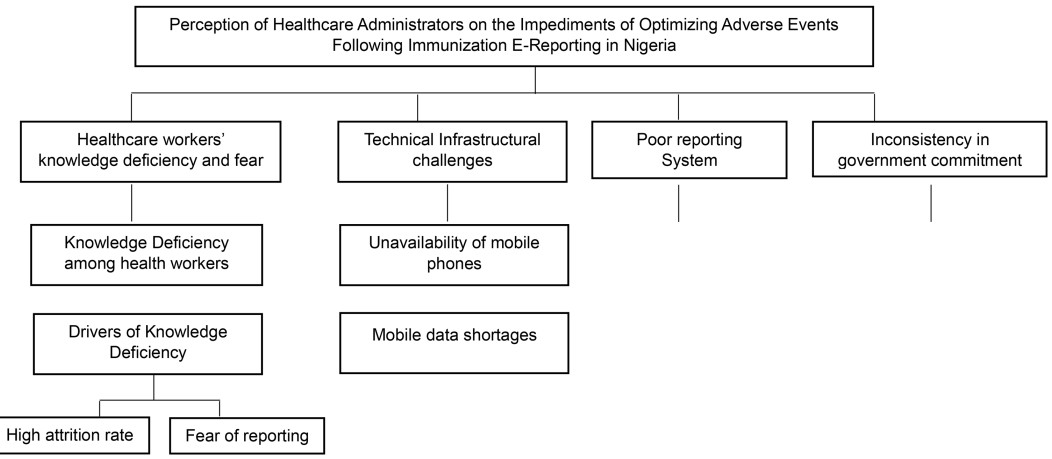

**Fig 1. Thematic structure analysis.**

(I) *High Attrition Rate*

Participants emphasized that knowledge deficiency remains extant in health facilities because of the high attrition rates that pervade the Nigerian health sector. Specifically, the exponential attrition rate in the health industry is often common among professional healthcare workers

*With the exodus of healthcare workers out of the country, you'll find that the people you trained might have left the facility, and so we might need to conduct another training course again (KII/National Level/2022).*

(II) *Fear of Reporting*

Complementing the lack of knowledge of AEFI was the fear of reporting AEFI among healthcare workers. This was propelled by misinformed notions that healthcare workers held based on reporting an AEFI event. This was evident in the enunciation of a few respondents that:

*They don't want anybody to say you don't know your work or you were the cause of this thing, so instead they don't report (KII/ State Level/2022).*

*The issue is that we discovered some of them are feeling that once they report AEFI, it is going to be like they're trying to tarnish or to mar the work they are doing as regards immunization (KII/ State Level/2022)*

**Technical infrastructural challenges**

Following subsequent discussions with the participants, the study revealed that the second major barrier identified was outside the control of healthcare workers, and this revolved around technological issues in health facilities. These technological issues include the unavailability of mobile phones, and mobile data shortages

**Unavailability of mobile phones.** Participants expressed that key health stakeholders at the local government and facility levels lack the devices relevant to reporting AEFIs at their respective levels, despite being an integral component of successful reporting

*Some of our local government DSNOs don't have Android phones which they can use in reporting through DHIS2 or Med Safety so that has been a challenge (KII/ State Level/2022).*

*So, if we can actually have a way to provide this electronic device to our RI-providing facilities, this can help (KII/ State Level/2022).*

**Mobile data shortages.** Consequently, the majority of the study participants revealed the need for a steady monthly data allowance, based on its consistent shortages. A steady monthly allowance can help facilitate the reporting and sharing of electronic data on AEFI as required at the state and national levels. To this end, some participants expressed that:

*If there are data subscriptions, that can go a long way in supporting, because all the facilities do not have data that is given to their staff (KII/ State Level/2022)*

*If it's possible for those that key in the data to be sent at least 2,500 monthly directly to their phones as a data bundle, it will help (KII/ State Level/2022)*

## Poor reporting system

This alludes to barriers that revolve around the functionality of the e-reporting portal and the modality of reporting. The study participants aired their perceptions of limitations to the reporting system or structure in place, and this was evident in the emphasis made by some participants that:

*You know, before we were not reporting AEFI electronically because it was paper-based. But now, different people have been complaining about the portal that it will not load sometimes. Some will tell you that they cannot login, and so because of that they were unable to appropriately report it. The DSNOs often complain that they do not receive any data from the facilities because they keep having different complains(KII/ State Level/2022).*

*Now, that recently we are using electronic data collection tools, what I realized is that the link or the platform can't be accessed to report AEFI from COVID-19 and especially other vaccination exercises (KII/ State Level/2022).*

## Inconsistency in government commitment

This is the fourth theme identified in the study based on the participants' responses. Findings from the study revealed that participants are of the notion that the government and health authority are not entirely playing their roles as expected, to enhance an effective and efficient electronic reporting system within the immunization sector. Some participants were able to shed light on this when they emphasized that:

*Like now, some health facilities are very far from other health facilities, and we have to go there to get data from them, so we can report to the state level. We complained to the state people, and they said they would provide small money for us to buy petrol in our machines, but we did not hear anything again (KII/ State Level/2022)*

*A big problem is that the government keeps promising that they will provide logistics, but we have not seen that since they said it. But if there are logistics for us at the state level to move around and to sensitize them and tell them the importance of reporting, I think our reporting method will be improved (KII/ State Level/2022).*

## Discussion

Our study set out to document stakeholders' perspectives on the challenges of AEFI reporting, with emphasis on electronic reporting, with a view to appraising suggestions for the optimization of AEFI reporting in Nigeria. The study participants discussed several issues that may impact the healthcare workers' perception and optimization of e-reporting of AEFI in Nigeria across the following themes: (1) healthcare workers' knowledge deficiency and Fear (2) Technical infrastructural challenges, (3) Poor reporting system, and (4) Inconsistency in government commitment.

The study's findings revealed that knowledge deficiency and fear of reporting were significant challenges. Participants noted that the general lack of knowledge about reporting AEFIs in health facilities was driven by limited interest on the part of healthcare workers, as well as the "Japa Syndrome" (attrition of healthcare workers who exit overseas) that has plagued various industries across the country. Okunade and Awosusi [24], defined Japa as the deployment of migration strategies (whether regular or irregular) to reside outside the shores of the Nigerian territory. Our finding on inadequate knowledge about reporting corroborates the work of Omoleke et al [25], who found the existence of varied and suboptimal knowledge levels of healthcare providers on AEFI definitions and classifications. This implies the existence of a significant relationship between knowledge deficiency and low reporting of AEFI, which may yield negative implications for decision-making at the highest level. Our study findings are similar to the study of Mehmeti et al [26], who observed that healthcare workers' varying interpretations of what defines a reportable AEFI lead to underreporting.

Our study observed that the negligence of healthcare workers in reporting AEFIs can also be corrected by establishing training sessions intermittently for healthcare workers across health facilities to always reiterate the value of reporting AEFIs. In furtherance, Participants from our study emphasized that healthcare workers at the facility level are usually riddled with fear of being perceived as incompetent by their superior officers should they report AEFI cases. This alludes to the findings of Aborigo et al [17,27] who reported "fear of blame by supervisors" as one of many impediments to reporting AEFI. This indicates that healthcare workers are, to an extent, consumed by misconceptions that in turn deplete their accurate knowledge of AEFI events. Essentially, worries about generating a "diminishing return" value in the eyes of superiors at the LGA level result in "cognitive dissonance" of healthcare workers at the facility levels, who are directly involved with clients reporting AEFI.

Findings from our study also illuminated that technical infrastructural problem was another significant barrier to reporting AEFI events presented at the health facilities. Owing to this, issues revolving around the lack of smart mobile phones, and consistent shortage of data services culminated in this. Omoleke *et al* [11] also found infrastructural deficits as one of ten major challenges that impede the electronic reporting of AEFI at the facility level. This raises concern about the access to mobile network challenge in particular, which in this case, is not strictly limited to visiting healthcare workers/administrators (such as the LGA DSNO/LIO), but also healthcare workers domiciled in the various electoral wards where primary healthcare facilities are situated. In other words, technical challenges will result in either delayed reporting or no reporting of AEFI events in such areas.

The possession of mobile smartphones in the facilities will enable facilities to document AEFI events electronically, along with the extant paper-based format in all facilities. Although mobile data services are not required to operate the application on the phones, nevertheless, mobile data service is required to synchronize and send the documented event. This also calls for stakeholders at the national level to involve responsible authorities to engage mobile network providers to provide the necessary pieces of equipment that will generate network services in these localities.

Participants emphasized that a poor reporting system was another major barrier that e-reporting faced in terms of reporting AEFI events. This is similar to the assertion of Mulchandani *et al* [28], who, through their study findings, claimed that under-reporting is common when reporting tools are unavailable and if healthcare workers perceive it as not being part of their responsibilities. Challenges were particularly experienced in the DHIS 2, where a section on AEFI was incorporated. To the participants, there was no coordinated structure to reporting AEFIs electronically, based on what they experienced in reporting. Essentially, participants conceived that having the LGA DSNOs go around all health facilities in the LGA was not sustainable, as they believed in a future potential disruption of the adopted reporting system. Consequently, there may be concerns on the part of the DSNO, who is likely to find his/her mobility across the LGA very strenuous and may sometimes resort to either not necessarily completing the facilities documentation, based on distance, or not visiting many facilities out of reluctance, resulting in little to no available accurate data. This aligns with the findings of Lv et al [19], who reported finding other methods of reporting AEFI as complicated. Additionally, the study found that having the new platform reporting only AEFIs for COVID-19 was a form of deterrence to healthcare workers, which was also considered a limitation of the reporting system. Healthcare workers might be discouraged because they look forward to a rather integrated electronic platform for reporting not only AEFIs for COVID-19 but also for AEFIs around other forms of immunization services.

Finally, the study observed that inadequate commitment on the part of the government and appropriate health authorities within the state was another inhibitor for the e-reporting of AEFIs in several health facilities across the country. This shares similarity with the findings of Ampratwum et al [29] who reported that the technical advisory committee of the AEFI at the higher level did not commit to managing AEFI cases reported to them from the facilities. The study participants emphasized that healthcare workers at the LGA level, and sometimes at facility levels, had usually brought forward complaints to the state level, particularly concerning logistics on the part of LGA representatives. However, the state authorities have always made false promises to the LGA healthcare workers and never committed to the intentions they made to

 

the complaining group. This, to a large extent, has served as a form of demoralization to healthcare workers, particularly those who believed in the promises made to them. Therefore, reneging on promises made to healthcare workers is a derogatory demeanour from the government toward healthcare workers who possess high hopes in seeking redress to their problems, and putting their trust in the government's promises to help them address their needs, through the appropriate health authority. Additionally, having a transportation means (such as a motorcycle) at the LGA level eases the supervision process to the various health facilities within a particular LGA, thereby resulting in an intrinsic form of motivation for healthcare workers at the LGA level.

## Implications of study findings

Findings from our study have established salient points integral to the understanding of the challenges affecting electronic pharmacovigilance reporting on Adverse Events Following Immunization (AEFI) based on stakeholders' perspectives. On this note, participants' perception of knowledge deficiency and fear among healthcare workers implies that the knowledge gap among healthcare workers regarding AEFI reporting is a severe challenge. Specifically, the lack of reporting knowledge has significant implications for the lives of many individuals who may be affected by adverse events following their immunization exercises. Additionally, the lack of knowledge of reporting on the part of healthcare workers has indirect implications on policies associated with adverse events following immunizations, as only limited data will be available, which may not be sufficient to inform appropriate decisions.

Furthermore, the search for greener pastures, which is highly prevalent among Nigerians, especially in the healthcare industry, is preponderantly responsible for the lack of knowledge on AEFI reporting within health facilities in the country. This implies that training of healthcare workers representing health facilities cannot be limited to a one-off event, as recruitment, replacement, and reshuffling because of attrition would translate to the need for routinized training. By extension, fear among healthcare workers raise concerns about how their work and competency will be perceived should they report AEFI events that were presented in their health facilities. This may be hinged on the belief held by healthcare workers that reporting AEFI is a negative outcome of their hard work toward vaccination exercises. In other words, healthcare workers may perceive reporting AEFI as tarnishing their reputation or undermining their immunization efforts. However, conducting sensitization (informative and educative) programs about AEFI and AEFI reporting at the state level and emphasizing the importance of reporting can help improve the reporting culture. It is therefore pertinent to address the concern of fear amongst healthcare workers and promote a supportive environment that would encourage healthcare workers to report AEFI cases without concerns over judgment or negative consequences.

Consequently, the lack of mobile phones dedicated to health facilities significantly impacts the reporting rate of AEFIs. This challenge limits healthcare workers' reporting capacity, leading to underreporting and incomplete data on AEFIs. Delayed reporting and response sometimes compromise the timely investigation and risk assessment. This challenge increases the risk of data inaccuracies and inconsistencies, impacting on the reliability of AEFI data and compromising effective analysis and decision-making

Participants highlighted that in some cases, especially in situations where the AEFI event is presented to them in their respective facilities, they are usually confronted with the challenge of mobile data shortages to synchronize and send out the reported data. In other words, the provision of mobile network data to healthcare workers for reporting AEFI electronically will not only serve as a tool for executing their responsibilities, but also as an agent of extrinsic motivation to the healthcare workers. As such, mobile data could help them reach out to their co-workers, especially the DSNOs, via social media pages such as WhatsApp, especially when such platforms are adopted for official purposes, including but not limited to check-in meetings, and communication and social mobilization.

The study participants exuded their expectations of the government's commitment towards their welfare, with particular reference to logistics to encourage their visits to the various health facilities, not only for data evaluation but also for supervision and on-the-job mentoring support. This implies that the government not reneging but rather committing to their promises

to the healthcare workers will significantly influence the execution of job responsibilities among healthcare workers, which in turn will improve the efficiency of the reporting system at the grassroots level to the pinnacle of the reporting system. This will also play an indirect role in the decision-making process of managing and controlling adverse events following immunization and/or drug administration. Additionally, the government's commitment can also be strengthened through improved counterpart funding and ownership of health program implementation. This is because of the limited political interest in health program interventions, especially when it require the collaboration of the national and subnational levels.

### Study strengths and limitations

This study has successfully illuminated the perspective of health administrators across two key levels: the Federal and State levels. On the other hand, the lack of assessing the experiences of facility health workers who primarily deal with AEFI reporting (direct experiences) is a major limitation of the study. Consequently, our findings may have been affected by social desirability bias, as many of our respondents might have answered the questions in a way that they believed was socially acceptable or desirable, rather than reporting their true opinion

### Conclusion

Adverse events following immunization are usually an aftermath of immunization but are not necessarily the effect of the immunization exercise. Data has revealed that information about reported AEFI cases in the country is relatively low across the 36+1 states of the country. This is evident from stakeholders' perception of various forms of challenges that have been reportedly synonymous with electronic AEFI case reporting from the health facility, However, findings from this baseline assessment informed the intervention from local development and implementing partners to train healthcare workers across two levels (LGA and Facility based levels) on electronic reporting of AEFI cases. Despite this endeavor, challenges (not necessarily technological) still prevail in documenting AEFI cases, and hence remain a bane to informed decision-making at the highest level.

Additionally, like several other health interventions in the country, it is pertinent to have key stakeholders across all levels on board, and in some cases, ensure multi-sectoral collaboration to drive the implementation activities and limit challenges or barriers to successful implementation of the improved AEFI reporting system.

### Recommendation

Based on findings from this study, we recommend that stakeholders, particularly at the state level, engage in frequent sensitization of healthcare workers on the benefits of reporting AEFI cases through all appropriately established channels (paper-based and electronic) to ensure that comprehensive data are available to make decisions at the right time, and in the right way. Additionally, stakeholders at the state level should also take responsibility for refresher training, and the provision of necessary tools for healthcare workers. Furthermore, training for healthcare workers (beyond AEFI reporting) should be institutionalized through policy and support from the national government with adequate resources across all health interventions. Stakeholders at the national level can support the sub-national, while also providing guidance and oversight functions to facilitate and ensure appropriate conduct.

Beyond healthcare workers' training, there is a need for the government to initiate multisectoral collaborations, especially with network providers, that can create a pathway for addressing network service provision challenges. Such a partnership would not only be beneficial to health workers, but also to community members in the vicinity of the facility. This partnership can also be leveraged to provide internet services at a limited cost for special phone numbers that can be registered to facilities only. There is also a need for the government to create a routine diagnosis for the AEFI reporting platforms that are controlled by the government (such as Med Safety App) and also connect such to other global databases that may be in use (like DHIS2). Finally, it is imperative on the part of the government to keep to the promises made to the public, including the healthcare workers, as this would foster political trust between the people and the government.

## Acknowledgments

We would like to acknowledge the Bill and Melinda Gates Foundation for providing the support and funding for the project intervention on strengthening the surveillance system of adverse events following immunization in Nigeria, through our organization (Sydani Group, Nigeria). We also want to extend our appreciation to the National Primary Healthcare Development Agency (NPHCDA) Board and the State Primary Healthcare Development Agency (SPHCDA) Board of every state that the Sydani team supported during the project phases.

## Author contributions

**Conceptualization:** Oluwafisayo Azeez Ayodeji, Euphemia Chigekwu Agomuo.

**Data curation:** Saheed Dipo Isiaka, Genevieve Ozioko, Euphemia Chigekwu Agomuo, Irene Odira Okoye, Precious Iyayi, Victor Daniel.

**Funding acquisition:** Sidney Sampson.

**Methodology:** Saheed Dipo Isiaka.

**Project administration:** Grace Fubara Erekosima, Folake Oni, Ahmed Rufai Garba.

**Resources:** Sidney Sampson.

**Supervision:** Grace Fubara Erekosima, Oluchi Bassey.

**Writing – original draft:** Saheed Dipo Isiaka, Euphemia Chigekwu Agomuo.

**Writing – review & editing:** Stephen Olabode Asaolu, Olugbemisola Wuraola Samuel, Oluwafisayo Azeez Ayodeji.

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
