## [Decision Letter · Decision Letter 0]

15 Nov 2024

PONE-D-24-32799
Perception of Healthcare Administrators on the Impediments of Optimizing Adverse Events Following Immunization E-Reporting in Nigeria

Dear Dr. Isiaka,

Thank you for submitting your manuscript to PLOS ONE. After careful consideration, we feel that it has merit but does not fully meet PLOS ONE’s publication criteria as it currently stands. Therefore, we invite you to submit a revised version of the manuscript that addresses the points raised during the review process.

We look forward to receiving your revised manuscript.

Kind regards,

Surya Bahadur Parajuli, MD

Academic Editor

PLOS ONE

1. When submitting your revision, we need you to address these additional requirements.
 
Please ensure that your manuscript meets PLOS ONE's style requirements, including those for file naming. The PLOS ONE style templates can be found at 
https://journals.plos.org/plosone/s/file?id=wjVg/PLOSOne_formatting_sample_main_body.pdf and 
https://journals.plos.org/plosone/s/file?id=ba62/PLOSOne_formatting_sample_title_authors_affiliations.pdf
 
2. In the online submission form, you indicated that [The data supporting the findings will be available from the corresponding author upon request. Requests will be examined and considered on a case-by-case basis.]. 
 
All PLOS journals now require all data underlying the findings described in their manuscript to be freely available to other researchers, either 1. In a public repository, 2. Within the manuscript itself, or 3. Uploaded as supplementary information.
 
This policy applies to all data except where public deposition would breach compliance with the protocol approved by your research ethics board. If your data cannot be made publicly available for ethical or legal reasons (e.g., public availability would compromise patient privacy), please explain your reasons on resubmission and your exemption request will be escalated for approval.
 
3. Your ethics statement should only appear in the Methods section of your manuscript. If your ethics statement is written in any section besides the Methods, please move it to the Methods section and delete it from any other section. Please ensure that your ethics statement is included in your manuscript, as the ethics statement entered into the online submission form will not be published alongside your manuscript.
 
4. We note that Figure 1 in your submission contain [map/satellite] images which may be copyrighted. All PLOS content is published under the Creative Commons Attribution License (CC BY 4.0), which means that the manuscript, images, and Supporting Information files will be freely available online, and any third party is permitted to access, download, copy, distribute, and use these materials in any way, even commercially, with proper attribution. For these reasons, we cannot publish previously copyrighted maps or satellite images created using proprietary data, such as Google software (Google Maps, Street View, and Earth). For more information, see our copyright guidelines: http://journals.plos.org/plosone/s/licenses-and-copyright.
 
We require you to either (1) present written permission from the copyright holder to publish these figures specifically under the CC BY 4.0 license, or (2) remove the figures from your submission:
 
1. You may seek permission from the original copyright holder of Figure 1 to publish the content specifically under the CC BY 4.0 license.  
 
We recommend that you contact the original copyright holder with the Content Permission Form (http://journals.plos.org/plosone/s/file?id=7c09/content-permission-form.pdf) and the following text:
“I request permission for the open-access journal PLOS ONE to publish XXX under the Creative Commons Attribution License (CCAL) CC BY 4.0 (http://creativecommons.org/licenses/by/4.0/). Please be aware that this license allows unrestricted use and distribution, even commercially, by third parties. Please reply and provide explicit written permission to publish XXX under a CC BY license and complete the attached form.”
 
Please upload the completed Content Permission Form or other proof of granted permissions as an ""Other"" file with your submission.
 
In the figure caption of the copyrighted figure, please include the following text: “Reprinted from [ref] under a CC BY license, with permission from [name of publisher], original copyright [original copyright year].”
 
2. If you are unable to obtain permission from the original copyright holder to publish these figures under the CC BY 4.0 license or if the copyright holder’s requirements are incompatible with the CC BY 4.0 license, please either i) remove the figure or ii) supply a replacement figure that complies with the CC BY 4.0 license. Please check copyright information on all replacement figures and update the figure caption with source information. If applicable, please specify in the figure caption text when a figure is similar but not identical to the original image and is therefore for illustrative purposes only.
The following resources for replacing copyrighted map figures may be helpful:
 
USGS National Map Viewer (public domain): http://viewer.nationalmap.gov/viewer/
The Gateway to Astronaut Photography of Earth (public domain): http://eol.jsc.nasa.gov/sseop/clickmap/
Maps at the CIA (public domain): https://www.cia.gov/library/publications/the-world-factbook/index.html and https://www.cia.gov/library/publications/cia-maps-publications/index.html
NASA Earth Observatory (public domain): http://earthobservatory.nasa.gov/
Landsat: http://landsat.visibleearth.nasa.gov/
USGS EROS (Earth Resources Observatory and Science (EROS) Center) (public domain): http://eros.usgs.gov/#
Natural Earth (public domain): http://www.naturalearthdata.com/

**Reviewers' comments:**

**Comments to the Author**

1. Is the manuscript technically sound, and do the data support the conclusions?

Reviewer #1: Partly

Reviewer #2: Yes

2. Has the statistical analysis been performed appropriately and rigorously? 

Reviewer #1: Yes

Reviewer #2: Yes

3. Have the authors made all data underlying the findings in their manuscript fully available?

Reviewer #1: Yes

Reviewer #2: Yes

4. Is the manuscript presented in an intelligible fashion and written in standard English?

Reviewer #1: Yes

Reviewer #2: Yes

5. Review Comments to the Author

Reviewer #1: Clarity and Structure:

The manuscript would benefit from clearer organization, particularly in the results section. Subheadings for each theme are helpful, but consider summarizing key findings at the beginning of each section to guide readers.

Thematic Analysis:

While the thematic categories are well-defined, the transitions between themes could be smoother. Consider providing a brief summary of how each theme relates to the overall findings, highlighting interconnections.

Literature Integration:

Integrate more references to existing literature on AEFI reporting challenges, within similar contexts. This would strengthen the manuscript by situating your findings within the broader research landscape.

Technical Details:

Provide more detail on the technological barriers mentioned. For example, discuss specific mobile technologies or platforms that could be improved or better utilized to enhance reporting.

Participant Diversity:

Consider discussing the diversity of respondents in more detail. Addressing the range of perspectives from different regions, roles, or levels of experience could enrich the analysis.

Cultural Context:

Incorporate a discussion of the cultural factors that may influence healthcare workers’ attitudes toward reporting AEFIs. Understanding local contexts can provide deeper insights into the identified barriers.

Training Programs:

Elaborate on existing training programs related to AEFI reporting. Discuss their effectiveness and any gaps that exist, which can inform recommendations for future training initiatives.

Government Role:

Provide a more nuanced discussion of the government's role in supporting AEFI reporting. Explore how policy changes or increased funding could facilitate improvements in reporting systems.

Follow-Up Mechanisms:

Address the need for follow-up mechanisms after reporting AEFIs. Discussing how feedback loops could enhance the reporting process might encourage more healthcare workers to engage in the system.

Technology Adoption:

Discuss barriers to technology adoption among healthcare workers in more depth. Exploring attitudes towards technology and previous experiences with reporting systems could provide a clearer picture of the challenges.

Visual Aids:

Consider including visual aids, such as charts or diagrams, to illustrate key themes and findings. Visual representation can enhance understanding and engagement with the material.

Stakeholder Engagement:

Highlight the importance of stakeholder engagement in the implementation of recommendations. Discuss how involving healthcare workers, policymakers, and community leaders can foster a more collaborative approach to addressing barriers.

Long-Term Strategies:

Propose long-term strategies for sustaining improvements in AEFI reporting. This could include establishing a continuous training framework or integrating reporting into regular health system assessments.

Case Studies:

If applicable, include brief case studies or examples of successful AEFI reporting systems from other regions or countries. This could provide a model for potential improvements in Nigeria.

Language and Terminology:

Ensure that technical terms and acronyms (like AEFI) are defined upon first use to make the manuscript accessible to a broader audience, including non-specialists.

Ethical Considerations:

Include a brief discussion of ethical considerations related to conducting interviews and reporting on sensitive issues in healthcare. This adds credibility to the research process.

Call to Action:

Conclude with a strong call to action that emphasizes the urgency of addressing the barriers identified. This can help mobilize stakeholders and create momentum for change.

Reviewer #2: The study appears to have technical soundness in several aspects but require to enhance methodological rigor and data presentation in order to alignment with publication standards. The choice of qualitative approach for exploring barriers to AEFI e-reporting allows for in depth insights for understanding nuances in reporting challenges. Information from stakeholders at both national and sub-national levels provides with a range of perspectives. The thematic analysis approach categories responses into well defined themes which is methodologically sound for qualitative studies. The sample size of 32 participants is reasonable for qualitative study. But, manuscript lack detailed descriptions of sampling method, interview protocols and specific data analysis processes. For instance, more information on participant recruitment criteria, sampling justification and interview guide content would improve the technical soundness of the study. The manuscript could also provide a clearer summary of data to support each theme, perhaps in a table. The discussion of how themes relate to existing literature is limited while incorporating a comparison with other studies strengthens the validity of findings. The conclusions align reasonably with findings, however, more direct references to participants' statements in relation to each conclusion would improve coherence and credibility. Ethical adherence is noted but explicitly discussing the informed consent process and any steps taken to reduced bias like training interviewers or using triangulations enhance transparency.

This manuscript is based on qualitative research and does not appear to involve quantitative data or statistical analysis. Instead, it employs thematic analysis, which is a standard approach for analyzing qualitative data. o While the manuscript mentions categorizing responses into themes, it would benefit from a more detailed description of the process, such as whether multiple coders were involved, how consistency in coding was ensured, or if any coding reliability measures were used. A clear explanation of coding process and any efforts to ensure consistency and examples of hoe the themes were derived from raw data improve rigor expected in qualitative analysis.

The manuscript is generally presented in an intelligible manner and uses standard English. It communicates the study's purpose, methodology, findings, and conclusions clearly. However, there are areas where the readability and clarity could be enhanced to meet publication standards fully. Adding clarifications for technical terms and more participant quotations would further enhance the intelligibility and engagement for readers. For Example, Some technical terms and concepts, such as “pharmacovigilance” and “thematic analysis,” could benefit from brief clarifications or definitions, particularly where these terms are first introduced.

While the data are technically available upon request, making de-identified data more accessible in a repository would strengthen the manuscript's adherence to open-access standards and facilitate reproducibility. If only summaries of the data (rather than full transcripts) can be shared, this should be specified along with instructions on how to request access. If ethical constraints prevent sharing full data, the authors should explain these limitations explicitly in the data availability statement and clarify that the data meet ethical standards regarding privacy and confidentiality.

6. PLOS authors have the option to publish the peer review history of their article (what does this mean?). If published, this will include your full peer review and any attached files.

Reviewer #1: **Yes: **Sajjad Ahmed Khan

Reviewer #2: No

---

## [Author Response · Author response to Decision Letter 1]

31 Jan 2025

While the data are technically available upon request, making de-identified data more accessible in a repository would strengthen the manuscript's adherence to open-access standards and facilitate reproducibility. If only summaries of the data (rather than full transcripts) can be shared, this should be specified along with instructions on how to request access. If ethical constraints prevent sharing full data, the authors should explain these limitations explicitly in the data availability statement and clarify that the data meet ethical standards regarding privacy and confidentiality

Response: Thank you for your observation. We share similar sentiments with you concerning making our data more open access. However, we would like to point out that the nature of our study makes this complicated. This manuscript is a sub-set of a landscape assessment conducted to optimize the AEFI surveillance and reporting system in Nigeria through a collaborative partnership among the Federal Ministry of Health, the National Primary Healthcare Development Agency, and Sydani Group, and funded by Bill and Melinda Gates Foundation (BMGF). Before the landscape assessment, an agreement was made with the Federal Ministry of Health, which is also the Institutional Review Board (IRB) custodian for all health-related research activities in Nigeria. The agreement includes a non-disclosure of facility-related information, which are heavily contained in the transcripts from the assessment. This is also why we could only write from a section of the assessment (because we believe it will be relevant to other developing countries). Providing the data as an open access will be considered a threat to national security by the two government health authorities in Nigeria. Hence, we would appreciate the journal permitting us only to provide our transcripts on a case-by-case request basis. Authors who were involved in the manuscript development and the project implementation (data access committee) will facilitate requests to share the data from the national-level stakeholders should the need arise (see contact details below).

Additionally, the funding body also highlighted in the project contract that all data obtained from the project are proprietary of the national health coordinating body and should only be utilized by our organization or any other third party upon agreement with the coordinating body.

The contact address that can be reached should the data be required is as follows

Organization name: Sydani Institute for Research and Innovation

Contact Person: Saheed Dipo Isiaka (email: saheed.isiaka@sydani.org)

Contact Person 2: Grace Erokosima (email: grace.erekosima@sydani.org)

Contact person 3: Genevieve Ozioko (email: genevieve.ozioko@sydani,org)

---

## [Decision Letter · Decision Letter 1]

11 Jul 2025

PONE-D-24-32799R1
Perception of Healthcare Administrators on the Impediments of Optimizing Adverse Events Following Immunization E-Reporting in Nigeria

PLOS ONE

Dear Dr. Isiaka,

Thank you for submitting your manuscript to PLOS ONE. After careful consideration, we feel that it has merit but does not fully meet PLOS ONE’s publication criteria as it currently stands. Therefore, we invite you to submit a revised version of the manuscript that addresses the points raised during the review process.

We look forward to receiving your revised manuscript.

Kind regards,

Edison Arwanire Mworozi, M.D

Academic Editor

PLOS ONE

Additional Editor Comments :

Please address further comments as indicated by reviewers and resubmit on time for expedition. Thank you.

Reviewers' comments:

Reviewer's Responses to Questions

**Comments to the Author**

Reviewer #2: All comments have been addressed

Reviewer #3: (No Response)

2. Is the manuscript technically sound, and do the data support the conclusions?

Reviewer #2: Yes

Reviewer #3: Partly

3. Has the statistical analysis been performed appropriately and rigorously? 

Reviewer #2: Yes

Reviewer #3: N/A

4. Have the authors made all data underlying the findings in their manuscript fully available?

Reviewer #2: Yes

Reviewer #3: Yes

5. Is the manuscript presented in an intelligible fashion and written in standard English?

Reviewer #2: Yes

Reviewer #3: Yes

6. Review Comments to the Author

Reviewer #2: (No Response)

Reviewer #3: This is an interesting qualitative study exploring barriers to AEFI reporting in Nigeria. The authors write that AEFIs are under-reported in Nigeria (though do not provide supporting evidence). To improve reporting, along with supporting vaccine safety surveillance and properly informing policy decisions, it is important to understand the reasons why AEFIs are not being reported, which may or may not be unique to this country. The authors identify four themes/potential challenges to AEFI reporting, though their only recommendation is health worker training. While overall, this manuscript is generally clearly written, there is some fine-tuning to do and details to iron out. For the Results section, the authors need to stick to the "facts" (strictly reporting) and in the Discussion, the authors might look at other systems that work emphasizing (what can be learned from other countries?), mention anything that is working well (has e-reporting improved AEFI reporting at all?), and really emphasize why this work is important.

Issues:

More important

• Check references: there are a number that need to be double checked for appropriateness/relevance. For example, references 2-4 (lines 40-45) do not appear to support the claims made in this paragraph. I was unable to see anything about preventing 2-3 million fatalities in reference 3. Reference 4 also does not seem very appropriate, and I am confident the authors can find a study that better supports the sentence. Additionally, references 1,3,14 are all the same – this needs to be corrected.

• Lines 65-72 include information on paper-based reporting. What is the process for e-reporting? Do the health workers report electronically through DHIS-2 and patients self-report through Med Safety? Or can health workers report through either app?

• In the Background section, you mention that AEFI reporting is low, below WHO reporting standards (for infants). It would be useful to provide statistics on this. How many AEFIs are reported annually in Nigeria? What is the reporting rate per population? Number or proportion of AEFIs reported electronically and number/proportion reported through the paper form would also be important to include here and gives readers a sense of the “problem”. Have rates increased/decreased/remained stable, including after implementing e-reporting? Additionally, are you interested in population-wide AEFIs, or only those associated with infant vaccines? If general population, the infant reporting standard is maybe less relevant.

• Lines 151-158: This paragraph gives some information on participants. It would be helpful to indicate which workers/positions are responsible for AEFI reporting and if possible, what they do. Is anyone directly involved in or responsible for reporting? I think it would be useful to briefly outline these participants’ roles in AEFI reporting.

• Lines 161-162: “Two states were chosen from each geo-political zone based on the highest reporting and lowest reporting indicators” – do highest and lowest reporting indicators refer to highest and lowest AEFI reporting? If not, what does this mean?

• Under “Data Collection”, can you include some examples of the questions asked or include a copy of the semi-structured interview questions in an appendix?

• Line 191: “…electronic pharmacovigilance systems defined by Agusoro et al (2018).” I think Agusoro may actually be Agoro et al [17]? Also, how was this paper/study used to make a codebook? Or what was used from the study? Was it the barriers these authors identified?

• Under “Results”, would it be feasible to number participants so the reader knows who is being quoted how many times? (The same person’s quotes could be used multiple times throughout the manuscript.)

• Under “Results”, are any of the participants responsible for reporting AEFIs? The quotes often use ‘they’, ‘those’ or ‘health workers’ instead of first person. It’s unclear if anyone interviewed has firsthand, direct experience and can speak to challenges directly. Otherwise, when I read the participant quotes, some of them sound as if they are passing on the blame to other people. It would be important to include the perspectives of those actually doing the reporting.

• Results for each of the four themes can be pared back a little bit – in some cases it seems like the reporting goes beyond the actual facts/information collected and may be expanding a little too much.

• The Discussion does not include a strengths and weaknesses section, which is an important omission.

• Lines 412-414: “Additionally, the study found that having the new platform report only AEFIs for COVID-19 was a form of deterrence to health workers , which was also considered a limitation of the reporting system.” This is confusing because it sounded like the reporting apps are not actually limited to just COVID and that this was a misconception related to lack of training/poor training. This should be clarified.

• Can you comment on the need to go from facility to facility for AEFI reporting? Would it be feasible to report AEFIs without having to visit each site? Is there anyway to streamline the process?

• One thing I would like to see in the Discussion is comparison of Nigeria’s AEFI reporting to other AEFI reporting systems. Are other countries doing the same thing? Who has a better system?

• Recommendation - I agree with this recommendation for continued/ongoing training. However, it sounds like the training hasn’t really improved reporting, or has it? Is this the only recommendation? Why not address the other 3 themes/barriers? Is there nothing else that can be improved or changed? If not, it would be good to mention that and why. What other steps or changes can help improve reporting? What parts of the system are working well?

More minor

• In the abstract, where it says “…but is probably not causally related to the vaccine” (line 15), I recommend changing this language. Instead of “probably not causally related” I would say something more along the lines of “may or may not be causally related” or even just “may not be causally…”

• In the abstract, under Objective (lines 19-20), I may include that you were identifying barriers to reporting, as that seems to be the focus of this manuscript. If changed, I would include this in the manuscript body as well.

• Also in the abstract, under Conclusion (lines 31-34), I would work on strengthening the conclusion because it currently reads somewhat vague.

• Line 62: spell out District Health Information Software 2 the first time and make sure Med Safety App has the proper spacing and capitalization

• Study Settings – While this information is interesting, it may not be necessary especially since none of it is used later in the manuscript - I could see it being included if there was discussion about how these factors or differences impact reporting rates or reporting culture or something along those lines/if later sections build off this information. This section could be shortened considerably.

• Line 99: Would be helpful to name the 12 states here in the text.

• Under Data Analysis, “The recorded files were equally transcribed verbatim in English language” (lines 185-186), what does “equally” mean here? Additionally, how were the interviews transcribed? Using AI software, a human, etc?

• Line 203: “…approval number NHREC/01/01/2007-19/08/2022” – is this study part of another study? When I tried to look up the NHREC approval number to find out more information, the only thing I can find is that “Patient-reported outcomes of adverse events after COVID-19 vaccination in Nigeria: A mixed methods study” published in 2024 has the same approval number.

• Line 216: Table 3 – this table does not seem to add much to the manuscript or reader understanding so I suggest either reporting these numbers in the text or trying to combine with Table 2.

• Health workers vs healthcare workers – I would choose one term and try to stick with it consistently throughout the manuscript. Also encourage you to pay attention to capitalization throughout.

• Line 261: what exactly is meant by “sensitization program”?

• Lines 294-296: “…healthcare workers are willing to report and share the AEFI event that has been presented to them in their respective facilities but are usually confronted with the challenge of lacking data to synchronize and send out the reported data.” Here, I would remove the bit about workers being willing to report if only they had data plans provided to them. Unless participants actually stated this, we don’t know whether they would be reporting if they had a mobile phone/data, especially given the other barriers to reporting, so this is along the lines of speculation.

• Lines 296-300: “In other words, the provision of mobile network data to healthcare workers for reporting AEFI electronically will not only serve as a tool for executing their responsibilities, but also as an agent of extrinsic motivation to the healthcare workers, as such mobile data could help them reach out to their co-workers, especially the DSNOs, via social media pages such as WhatsApp.:” I imagine the government/ministry does not want workers to use their work phones or work mobile plans to go on social media. If this would help them reach out to coworkers using a messenger app (such as WhatsApp) for work purposes, I think that would be a different story.

• Lines 363-364: “Participants enumerated that the general lack of knowledge of reporting AEFIs in health facilities was driven by negligence on the part of the healthcare workers…” My recommendation here would be tone down this language from ‘negligence’ to something else. This word sounds a little harsh and as though you are placing the blame on the workers when there are other factors at play as well.

• Lines 367-369: “This corroborates the work of Omoleke et al [19] who found the existence of varied and suboptimal knowledge levels of healthcare providers on AEFI definitions and classifications.” The previous sentence talks about japa and it’s unclear how this sentence relates to japa. You need to connect these two ideas better.

• Lines 430-431: “…and tend to over-rely on the government , with the notion that the government through the appropriate health authority is expected help them address their needs.” It strikes me as odd to say – who should they be looking to?

7. PLOS authors have the option to publish the peer review history of their article (what does this mean?). If published, this will include your full peer review and any attached files.

Reviewer #2: No

Reviewer #3: No

---

## [Author Response · Author response to Decision Letter 2]

8 Aug 2025

Thank you for your observation. However this has been addressed long ago. I am certain that yo are aware about this, but the platform keeps raising this question

IN the last email sent (request to edit), no point was raised about what to be included in the edit. However, I have followed the advice from the email sent earlier the same day. in the "file attachment component, I have reduced the number of files on the platform, leaving the previously submitted documents, and the most recent documents (submitted today) for editorial purposes. Kindly let me know if this is aaceptable or more clarification can be provided on what is expected of the authors.

---

## [Editor Report · Decision Letter 2]

12 Aug 2025

Perception of Healthcare Administrators on the Impediments of Optimizing Adverse Events Following Immunization E-Reporting in Nigeria

PONE-D-24-32799R2

Dear authors,

We’re pleased to inform you that your manuscript has been judged scientifically suitable for publication and will be formally accepted for publication once it meets all outstanding technical requirements.

Kind regards,

Edison Arwanire Mworozi, M.D

Academic Editor

PLOS ONE

---

## [Editor Report · Acceptance letter]

PONE-D-24-32799R2

PLOS ONE

Dear Dr. Isiaka,

I'm pleased to inform you that your manuscript has been deemed suitable for publication in PLOS ONE. Congratulations! Your manuscript is now being handed over to our production team.

Kind regards,

on behalf of

Professor Edison Arwanire Mworozi

Academic Editor

PLOS ONE